# Genome-Wide Meta-Analysis Identifies Variants in *DSCAM* and *PDLIM3* That Correlate with Efficacy Outcomes in Metastatic Renal Cell Carcinoma Patients Treated with Sunitinib

**DOI:** 10.3390/cancers14122838

**Published:** 2022-06-08

**Authors:** Meta H. M. Diekstra, Jesse J. Swen, Loes F. M. van der Zanden, Sita H. Vermeulen, Epie Boven, Ron H. J. Mathijssen, Koya Fukunaga, Taisei Mushiroda, Fumiya Hongo, Egbert Oosterwijk, Anne Cambon-Thomsen, Daniel Castellano, Achim Fritsch, Jesus Garcia Donas, Cristina Rodriguez-Antona, Rob Ruijtenbeek, Marius T. Radu, Tim Eisen, Kerstin Junker, Max Roessler, Ulrich Jaehde, Tsuneharu Miki, Stefan Böhringer, Michiaki Kubo, Lambertus A. L. M. Kiemeney, Henk-Jan Guchelaar

**Affiliations:** 1Department of Pharmacology, Princess Maxima Center for Pediatric Oncology, 3584 CS Utrecht, The Netherlands; 2Department of Clinical Pharmacy and Toxicology, Leiden University Medical Center, 2333 ZA Leiden, The Netherlands; j.j.swen@lumc.nl (J.J.S.); s.boehringer@lumc.nl (S.B.); h.j.guchelaar@lumc.nl (H.-J.G.); 3Radboud University Medical Center, 6525 GA Nijmegen, The Netherlands; loes.vanderzanden@radboudumc.nl (L.F.M.v.d.Z.); sita.vermeulen@radboudumc.nl (S.H.V.); bart.kiemeney@radboudumc.nl (L.A.L.M.K.); 4Department of Medical Oncology, Amsterdam UMC, Vrije Universiteit Amsterdam/Cancer Center Amsterdam, 1081 HV Amsterdam, The Netherlands; e.boven@amsterdamumc.nl; 5Department of Medical Oncology, Erasmus MC Cancer Institute, 3015 GD Rotterdam, The Netherlands; a.mathijssen@erasmusmc.nl; 6Core for Genomic Medicine, RIKEN Center for Integrative Medical Sciences, Yokohama 230-0045, Japan; koya.fukunaga@riken.jp (K.F.); mushiroda@riken.jp (T.M.); michiaki.kubo@riken.jp (M.K.); 7Department of Urology, Kyoto Prefectural University of Medicine, Kyoto 602-8566, Japan; fhongo@koto.kpu-m.ac.jp (F.H.); tmiki@koto.kpu-m.ac.jp (T.M.); 8Radboud Institute for Molecular Life Sciences, Radboud University Medical Center, 6525 GA Nijmegen, The Netherlands; egbert.oosterwijk@radboudumc.nl; 9CERPOP, Center of Epidemiology and Research in Population Health, Joint Unit 1295, Institut National de la Santé et de la Recherche Médicale (Inserm), Faculty of Medicine, Université de Toulouse, Université Toulouse III-Paul Sabatier, CEDEX, 31062 Toulouse, France; anne.cambon-thomsen@univ-tlse3.fr; 10Medical Oncology Department, I+12 Research Institute, (CiberOnc), Hospital Universitario 12 de Octubre, 28041 Madrid, Spain; cdanicas@hotmail.com; 11Institute of Pharmacy, Clinical Pharmacy, University of Bonn, 53117 Bonn, Germany; achim.fritsch@outlook.com (A.F.); u.jaehde@uni-bonn.de (U.J.); 12Medical Oncology, HM Hospitales—Centro Integral Oncológico HM Clara Campal, 28050 Madrid, Spain; jgarciadonas@gmail.com; 13Hereditary Endorine Cancer Group, Spanish National Cancer Research Center (CNIO) and Biomedical Network on Rare Diseases (CIBERER), 28029 Madrid, Spain; crodriguez@cnio.es; 14PamGene International B.V., 5211 HH ‘s-Hertogenbosch, The Netherlands; rru@genmab.com; 15University of Medicine and Pharmacy Carol Davila, 050474 Bucharest, Romania; tudoradu@gmail.com; 16Department of Oncology, Cambridge University Hospitals NHS Foundation Trust, Cambridge Biomedical Campus, Cambridge CB2 0SL, UK; tim.eisen@medschl.cam.ac.uk; 17Clinic of Urology and Paediatric Urology, Saarland University, 66424 Homburg, Germany; kerstin.junker@uniklinikum-saarland.de; 18CESAR Central Office, CESAR Central European Society for Anticancer Drug Research-EWIV, 1010 Vienna, Austria; max.roessler@cesar.or.at; 19Department of Biomedical Data Sciences, Leiden University Medical Center, 2333 ZA Leiden, The Netherlands

**Keywords:** genome wide association study, sunitinib, pharmacogenetics, metastatic renal cell carcinoma, clear cell renal cell carcinoma, single nucleotide polymorphism

## Abstract

**Simple Summary:**

The drug sunitinib is used in metastatic renal cell carcinoma, but patients respond very differently to this drug. To better tailor sunitinib treatment to the individual patient, clinically useful markers are needed. We explored the DNA of patients with metastatic renal cell cancer to detect variations that determine how a patient would respond to sunitinib treatment. We investigated >8 million genetic variants in large patient cohorts from Europe (*n* = 550) and Japan (*n* = 204) and found novel genetic variants in *PDLIM3* and *DSCAM* that are related to survival in sunitinib-treated patients. The mechanistic role of these variants in the action of sunitinib needs to be further explored to define its clinical potential. Our findings are a major step towards achieving personalized treatment for patients with metastatic renal cell carcinoma.

**Abstract:**

Individual response to sunitinib in metastatic renal cell carcinoma (mRCC) patients is highly variable. Earlier, sunitinib outcome was related to single nucleotide polymorphisms (SNPs) in *CYP3A5* and *ABCB1*. Our aim is to provide novel insights into biological mechanisms underlying sunitinib action. We included mRCC patients from the European EuroTARGET consortium (*n* = 550) and the RIKEN cohort in Japan (*n* = 204) which were analysed separately and in a meta-analysis of genome-wide association studies (GWAS). SNPs were tested for association with progression-free survival (PFS) and overall survival (OS) using Cox regression. Summary statistics were combined using a fixed effect meta-analysis. SNP rs28520013 in *PDLIM3* and the correlated SNPs rs2205096 and rs111356738 both in *DSCAM*, showed genome-wide significance (*p* < 5 × 10^−8^) with PFS and OS in the meta-analysis. The variant T-allele of rs28520013 associated with an inferior PFS of 5.1 months compared to 12.5 months in non-carriers (*p* = 4.02 × 10^−10^, HR = 7.26). T-allele carriers of rs28520013 showed an inferior OS of 6.9 months versus 30.2 months in non-carriers (*p* = 1.62 × 10^−8^, HR = 5.96). In this GWAS we identified novel genetic variants in *PDLIM3* and *DSCAM* that impact PFS and OS in mRCC patients receiving sunitinib. The underlying link between the identified genes and the molecular mechanisms of sunitinib action needs to be elucidated.

## 1. Introduction

Sunitinib is a multi-targeted tyrosine kinase inhibitor (TKI) and is, next to immunotherapy, an important component in the treatment of patients with metastatic renal cell carcinoma (mRCC) [1,2,3]. Treatment in mRCC patients is started based on the risk group which is determined by clinical and pathological characteristics, but does not yet provide an adequate prediction of treatment outcome. The most common histological subtype is clear cell renal cell carcinoma and occurs in about 75% of cases [4]. And although sunitinib is effective in clear cell mRCC, there is large inter-individual variability in the response to sunitinib regarding both side-effects and efficacy. One-third of patients require a dose reduction and up to 20% show no clinical response to sunitinib [1,2]. Biomarkers that enable prediction of individual response to sunitinib are imperative [5]. In previous studies, the candidate gene approach was used to test single nucleotide polymorphisms (SNPs) in genes related to the pharmacokinetics (PK) and pharmacodynamics of sunitinib. SNPs in *CYP1A1*, *CYP3A5*, *CYP3A4*, *NR1I2*, *NR1I3*, *ABCB1*, *ABCG2*, *VEGF-A*, *VEGF-R1*, *VEGFR2*, *VEGF-R3*, *FGF-R2*, *FLT3*, *eNOS*, *UGT1A1* and *IL8* were significantly associated with one or more of the following endpoints: toxicity, dose reduction, clearance, drug exposure, best objective response, progression-free survival (PFS) or overall survival (OS) [1,2,5,6,7,8,9,10,11,12,13,14,15,16]. Only genetic polymorphisms in *CYP3A5* (rs776746) and *ABCB1* (rs1128503, rs2032582, rs1045642) were replicated for association with a twofold increase in the need for dose reductions and with PFS, respectively [17]. Germline genetic variants are, therefore, considered to be major contributors to differences in response to sunitinib. However, sample sizes in most candidate gene studies were limited (up to 350 subjects), replication is often lacking, and inconsistent endpoint definitions were used making it difficult to draw firm conclusions [1,2,5,6,7,8,9,10,11,12,13,14,15,16].

In earlier genome-wide association studies (GWASs) in other diseases, genetic variants were associated with drug response or adverse events of commonly used drugs such as simvastatin, warfarin, and flucloxacillin, and this affected clinical practice. However, few GWAS data are available for response to anticancer agents [18]. The hypothesis-free approach of a GWAS can provide novel insights into biological mechanisms underlying sunitinib action. Yet, any GWAS in cancer remains challenging because of the need of large sample sizes, and ideally a validation in an independent cohort [18].

GWAS data on response and toxicity in mRCC patients treated with pazopanib (*n* = 744) or sunitinib (*n* = 355) have been presented in 2014 [19,20]. SNPs in *IL2RA* and *LRRC2* were associated with efficacy endpoints at a threshold of *p* ≤ 5 × 10^−7^, just falling short of genome-wide significance (i.e., *p* ≤ 5.0 × 10^−8^) [15,19]. Intronic variants in *LOXL2* and *ENTPD4* were associated with PFS, OS and best response as a combined efficacy endpoint (*p* = 1.7 × 10^−8^). However, for individual efficacy endpoints no genetic variants were shown to be associated at GWAS significance level [19,20].

We established the “TArgeted therapy in Renal cell cancer: GEnetic and Tumour related biomarkers for response and toxicity” (EuroTARGET) consortium to search for sunitinib biomarkers [21]. Here, we report the results of the largest meta-GWAS for sunitinib with the aim to identify germline genetic variants that associate with sunitinib efficacy in a cohort of mRCC patients as recruited by the EuroTARGET consortium, and a cohort available from the RIKEN Centre in Japan.

## 2. Materials and Methods

### 2.1. Patient Cohorts

Within the EuroTARGET project, a collection of blood samples and tissue material of prospectively included patients is available as well as a collection of stored samples of ‘historical patients’ enrolled in earlier studies [21]. Patients were recruited at participating centres in the Netherlands, the United Kingdom, Iceland, Germany, Romania and Spain. The enrolment of patients in this study occurred from 2005 until 2015 [21,22]. Inclusion criteria for the EuroTARGET GWAS were a histologically confirmed diagnosis of mRCC, the use of sunitinib as a first TKI, the availability of germline DNA, and recorded clinical data on PFS and OS.

For the RIKEN cohort, a total of 219 mRCC patients treated with sunitinib were recruited from 15 Japanese medical institutes. PFS and OS data were available for 204 individuals. All patient data were collected at the Centre for Integrative Medical Sciences at the laboratory for pharmacogenomics of the research institution RIKEN, in Tokyo [23]. For this cohort, GWAS summary statistics (effect sizes, standard errors) were used in the current analyses.

### 2.2. Patient Selection

Patients from the EuroTARGET cohort were considered eligible for our genetic association analysis if genotyping data could be obtained, if the start date of sunitinib and a follow-up date after the start date were available, and if the follow-up after start of sunitinib until study end was more than 24 weeks. To enable informative future analyses, we only focus on the subset of patients for whom outcome could be assessed for at least 6 months (24 weeks). Patients were only included if no TKI or TKI-like anti-tumour treatment was given prior to sunitinib (Appendix A). Patients with a clear cell histological subtype as well as those with unknown histology were included, while other histological subtypes were excluded [21]. For the ‘historical patients’ whole blood, serum or plasma material had been obtained. For each of the prospective patients, two 10 mL blood samples were collected. One of the two samples was stored in a central biobank in the Netherlands, and the other was used for genotyping. Genotyping of retrospective and prospective samples was performed at deCODE Genetics in Reykjavik, Iceland.

The EuroTARGET study has been approved by the local research ethics committees of all participating centres and all patients gave their written informed consent. For subjects of the Dutch SUTOX consortium (the ‘historical patients’ in the EuroTARGET cohort), DNA samples were anonymized by a third party according to the instructions stated in the Codes for Proper Use and Proper Conduct in the Self-Regulatory Codes of Conduct (www.federa.org (accessed on 24 March 2020)) [8,21].

### 2.3. Clinical Data Collection

For the analyses, we used the clinical EuroTARGET database version from October 2017. Clinical data from all patients of the EuroTARGET cohort were collected by medical file review and entered in web-based electronic case record forms (eCRFs) (Appendix A). Data included demographic information, baseline clinical characteristics, treatment lines, drug toxicities (Common Terminology Criteria for Adverse Events [CTCAE version 4.0]), tumour response (i.e., complete remission, partial remission, stable disease, or progressive disease), and death. Tumour response was defined according to RECIST version 1.1 and based on patient evaluation by local caregivers as given in the radiology report or medical record (no review of imaging) [21].

### 2.4. Genotyping and Quality Control (QC)

EuroTARGET: Genotyping of the EuroTARGET samples was conducted at deCODE genetics (Reykjavik, Iceland). Germline DNA isolated from whole blood with the Chemagic Blood kit (PerkinElmer, Waltham, Massachusetts, United States) was used for SNP genotyping on the Illumina Human OmniExpress BeadChips 12v1-1, 24v1-0, and 24v1-1 (Illumina, San Diego, CA, USA) [24]. Quality control (QC) checks for Eurotarget were performed using software R version 3.2.3 [25] and PLINK software, version 1.07 [26]. Individuals were excluded from analyses based on an individual genotype call rate <97%, gender mismatch between reported and estimated sex based on genotypes of the X-chromosome, or excess of heterozygous genotypes (i.e., inbreeding statistic of F > 0.1). Genetic markers were excluded based on a SNP call rate <97%, minor allele frequency (MAF) <1%, and a *p*-value ≤ 10^−7^ for the Hardy-Weinberg equilibrium (HWE) goodness-of-fit test. After exclusion of individuals and markers in these marginal QCs, the remaining set was used for integrative QC assessment. To evaluate the possibility of population stratification or outliers, multidimensional scaling (MDS) analysis was performed and pairwise identity by state (IBS) statistics was calculated to assess duplicates. Both MDS and IBS were computed using PLINK [26]. Individuals that were identified as outliers were excluded. SNP imputation was performed using shapeit and impute2 with default parameters and the reference panel 1000 genomes build version 3 with total ‘cosmopolitan’ set of individuals [27,28,29].

RIKEN: QC procedures for RIKEN were performed prior to association analyses and similar to those performed for the EuroTARGET cohort. Individuals were excluded based on an individual genotype call rate ≤98%. However, no exclusion was carried out based on excess of heterozygous genotypes. For population stratification, a principal components analysis (PCA) was executed in which all the 204 subjects were reported to be Japanese. Genetic markers were excluded based on a SNP call rate <97%, a MAF <1%, and a *p*-value ≤ 10^−7^ for HWE.

### 2.5. Genetic Association Analysis

Cox-regression analyses were performed for PFS and OS correcting for covariates. For the meta-analysis, data of the EuroTARGET cohort were combined with data from the RIKEN cohort [23]. GWAS summary statistics of both cohorts were combined using a fixed effect meta-analysis (R-package metafor). When summary statistics of only one study were available, this result was used in the combined analysis.

Eurotarget cohort: For each SNP, genotypes were tested for association with efficacy outcomes using Cox proportional hazard regression analysis. The primary efficacy endpoint was PFS and the secondary outcome was OS. PFS was defined as the time in months between the first day of sunitinib treatment and the date of progressive disease (PD) according to RECIST version 1.0 and 1.1. If no PD was observed, PFS was censored at the time of the last follow-up or death. OS was defined as the time in months between the first day of sunitinib treatment and the date of death or the date at which the patient was last known to be alive. SNPs and additional covariates age at start of sunitinib treatment, gender, country, and Heng prognostic risk group (favourable, intermediate or poor) were included in the cox regression for outcomes PFS and OS using an additive genetic model [30]. Statistical analyses were performed in R statistics version 3.2.3, using base package survival to evaluate Cox regressions. To impute missing values for the Heng variables (i.e., WHO performance status, haemoglobin, neutrophil count, thrombocytes, calcium, and time from diagnosis until start of sunitinib), R-package mice was used with 100 imputations. From these 100 imputations, the most likely Heng score was imputed as a single imputation. Associations with a *p*-value ≤ 5 × 10^−8^ were considered genome-wide significant. Associations between *p* = 5 × 10^−8^ and *p* = 5 × 10^−7^ were considered suggestive. Post association QC was performed by visual inspection of *p*-values in the Quantile-Quantile (QQ) plots and computation of the inflation factor λ.

RIKEN cohort: GWAS summary statistics (HRs and standard errors) in the RIKEN cohort were tested for association on PFS and OS. Association analyses were adjusted for age, gender, ECOG performance and RCC histology [23].

## 3. Results

### 3.1. Patients and Genetic Data

Clinical data were available for 713 sunitinib treated patients in the EuroTARGET cohort. In total, 85 patients were removed from the analysis because their blood sample contained insufficient DNA. The remaining 628 patients entered the QC prior to association analyses (Figure 1) [21]. The observed individual genotype call rates varied between 99.2 to 100% and met the quality criteria. Based on further quality control steps, 24 patients were excluded from analysis. A sample mix-up rate of 0.2% was observed resulting in the removal of two patients as well as 20 individuals recognized as outliers by the multi-dimensional scaling (MDS) analysis (Appendix A). The mean inbreeding coefficient was F = 0.01 (95% CI: −0.02, 0.04), leading to the exclusion of two patients with an inbreeding coefficient of F > 0.1. Pairwise IBS scores are plotted in Appendix A. For efficacy analysis, 54 patients with a non-clear cell subtype of RCC were excluded. This resulted in a total of 550 mRCC patients for whom information on efficacy data and genotypes were available, and these were included in the EuroTARGET cohort analysis (Figure 1). The starting dose of sunitinib was 50 mg/day in most patients (*n* = 482, 87.6%) in a 4-weeks-on/2-weeks-off schedule. Patient characteristics for both the EuroTARGET cohort and the RIKEN cohort are shown in Table 1.

For the EuroTARGET cohort, median follow-up times of PFS and OS were 7.6 months (range: 3 days–112.5 months) and 17.0 months (range: 9 days–112.5 months), respectively. The quality criteria for statistical analysis were met for 679,324 SNPs based on measured genotypes (Appendix A). This number was supplemented using SNP imputation procedures using the 1000 Genomes data as reference panel [29]. A total of 8,148,675 SNPs with a minor allele frequency (MAF) of >2% were analysed for the EuroTARGET cohort.

For the RIKEN cohort, median follow-up times of PFS and OS were 10.6 months (range: 0.4–69.7 months) and 12.3 months (range: 0.5–69.7 months), respectively. For the 204 individuals from the RIKEN cohort, QC procedures were performed prior to association analyses. A total of 5,518,066 SNPs with a minor allele frequency (MAF) of >2% were analysed for the RIKEN cohort.

Duration of prior drug treatment refers to the number of months that a patient has received an antitumor treatment prior to start with sunitinib that is not a TKI or TKI-like drug as mentioned in Appendix A, because these patients have already been excluded from analyses.

### 3.2. Genetic Association Analysis

For the meta-analysis, results of statistically significant or suggestive association with PFS or OS are presented in Table 2 and Table 3, respectively. The GWAS results for the meta-analysis and separate cohorts are visualized in Manhattan plots in Figure 2 and Figure 3. The associated SNPs in the meta-analysis come from imputed SNPs in the EuroTARGET cohort and are not present in the Japanese dataset. The annotation and interpretation of our findings therefore apply only to European populations. In the meta-analysis, SNP rs28520013 in *PDLIM3* (*p* = 4.02 × 10^−10^, HR = 7.26), and SNPs rs2205096 (*p* = 5.60 × 10^−9^, HR = 2.5) and rs111356738 (*p* = 4.77 × 10^−8^, HR = 2.51), both in *DSCAM*, were significantly associated with PFS. Most of the SNPs in *DSCAM* show high levels of linkage disequilibrium (R^2^ > 0.8) (Appendix A) suggesting a single causal variant at this locus. However, for SNPs rs2205096 and rs111356738 in *DSCAM* the R^2^ is 0.694. Furthermore, suggestive association with PFS was found for 7 SNPs in *DSCAM, DNASE1L3*, *CALN1*, *LIMCH1*, or in proximity of *LOXL4*, *PYROXD2*, *HPS1*, and *HPSE2*. Results from the meta-analysis on OS, showed that SNP rs28520013 in *PDLIM3* (*p* = 1.62 × 10^−8^, HR = 5.96) was the statistically strongest finding. Additionally, SNPs in *CACNA2D3*, *ART1*, *DAB1*, *LIMCH1*, and SNPs close to *PTPRD* were suggestively associated with OS (*p*-values between 6.75 × 10^−8^ and 4.36 × 10^−7^). Median PFS for non-carriers of the *PDLIM3* variant in rs28520013 was 12.5 months (95% CI: 10.9–14.9) versus only 5.1 months (95% CI: 1.5–6.2) for variant T-allele carriers (*p* = 4.02 × 10^−10^, HR = 7.26). For OS, the median was 30.2 months (95% CI: 27.4–33.4) for non-carriers of rs28520013 versus only 6.9 months (95% CI: 5.5–27.6) for the variant T-allele carriers (*p* = 1.62 × 10^−8^, HR = 5.96). The variant A-allele of SNPs rs2205096 in *DSCAM* was associated with an inferior PFS compared to non-carriers (*p* = 5.60 × 10^−9^, HR = 2.50) with a median PFS of 1.5 months for the AA genotype, 6.8 months for AT, and 12.6 months for the TT genotype. Also, the variant A-allele of rs111356738 in *DSCAM* was associated with an inferior PFS versus non-carriers (*p* = 4.77 × 10^−8^, HR = 2.51) with a median PFS of 1.5 months for the AA genotype, 5.6 months for AT, and 12.7 months for the TT genotype.

Significant or suggestive associations with PFS and OS in the RIKEN and EuroTARGET cohort separately are presented in Appendix A. The QQ plots from the EuroTARGET cohort are shown in Appendix A, with inflation factors of 1.08 and 1.05.

## 4. Discussion

This GWAS identified novel germline DNA variants in *PDLIM3* and *DSCAM* for PFS and OS in sunitinib treated mRCC patients. The most significant finding with a large effect size on both PFS and OS was found for SNP rs28520013 in *PDLIM3*. Variants in *PDLIM3* and *DSCAM* may well modify p21-associated kinases (PAKs) activity and influence the PI3K/AKT signalling pathway and thereby modify drug resistance and decrease sunitinib efficacy through NF-κB/IL-6 activation (Appendix A) [31,32,33,34,35,36,37,38,39]. Yet, this mechanism is hypothetical, based on previous literature findings. The underlying link between the identified novel candidate genes and the molecular mechanisms of sunitinib action remains to be elucidated.

This is the largest pharmacogenetic association study for outcome in sunitinib-treated mRCC patients to date. Our current knowledge on the pharmacology of sunitinib and its impact on efficacy is not represented in this GWAS, and thus may involve other mechanisms that contribute to efficacy. The suggestively associated SNP rs595883 with PFS in the meta-analysis was located close to *LOXL4*, which is in the same lysyl oxidase-like gene family as the earlier reported *LOXL2* gene in the GWAS by Motzer et al. [19]. Furthermore, both the identified SNP associations for *DSCAM*, *PDLIM3*, and *CACNA2D3* in this study as well as the earlier reported SNP associations for *IL2RA* in the GWAS of Motzer et al. may act via the PI3K/AKT pathway [19]. As opposed to earlier studies, we aimed to confirm our findings in a meta-analysis, which emphasizes robustness. However, rs28520013 in *PDLIM3* was analysed only in the EuroTARGET cohort. On the other hand, rs28520013 was associated with both PFS and OS lending support to its importance.

The associated SNPs in *PDLIM3* and in *DSCAM* are intronic variants. *PDLIM3* encodes the PDLIM3 protein from the LIM family and contains a PDZ and LIM domain. This protein is found in muscle cells and polymorphisms in *PDLIM3* were earlier associated with systolic blood pressure and an increased risk of cardiomyopathy [31]. It is assumed that PDLIM proteins negatively regulate NF-κB by inhibition of p65 activation. PDLIM1 deficiency in mice results in augmented production of cytokines [32]. Furthermore, a subfamily of the PDZ-LIM proteins (LIMK1) possibly interacts with receptor tyrosine kinases [33]. The *DSCAM* gene is located on chromosome 21q22.2-q22.3 in the Down Syndrome Critical Region (DSCR) and encodes the Down Syndrome Cell Adhesion Molecule (DSCAM), an immunoglobulin-superfamily member adhesion molecule. DSCAM regulates the function of p21-associated kinases (PAKs) that belong to the serine/threonine intracellular protein kinases [34,35,36]. Furthermore, expression of *CACNA2D3* was reported to enhance chemosensitivity to cisplatin therapy by inducing Ca^2+^-mediated apoptosis and blocking the PI3K/AKT signalling pathways [37]. LOXL4 expression was related to an improved OS in liver cancer patients with wild-type p53 tumours [38]. A theory that fits with our findings is ‘phenoconversion’, in which an increased cytokine expression could inhibit the metabolism of sunitinib and thereby increase actual drug exposure, resulting in favourable PFS and OS outcomes [39].

Differences in associated loci or SNPs between the two cohorts may be explained by possible ethnicity-specific effects, including differences in MAFs. Also, the number of analysed SNPs and hence genome coverage differed between the two cohorts. When comparing SNPs across the cohorts allowing for some physical distance, more overlap can be identified in an explorative way. While other approaches are possible, such as gene-based analyses, they were outside the scope of this study. The frequency of the variant T-allele of rs28520013 in *PDLIM3* in this study was 2.3% while the frequency reported in Asians in the NCBI database is 0.1% [40]. This suggests that an Asian population can hardly contribute to this SNP association. For the variant A-allele of rs2205096 in *DSCAM* the MAF reported in this study is 6.3%, whereas the reported MAF for Asians in the NCBI database is 36% [40]. SNPs rs2205096 and rs111356738 in *DSCAM* are moderately to highly correlated (R^2^ = 0.7) and other observed SNPs in *DSCAM* were strongly correlated with rs2205096 (R^2^ > 0.8). Hence, SNPs in *DSCAM* are likely not to act independently on PFS, and sequence analyses and possibly experimental systems would be needed to determine the actual causal variant.

A clear distinction between the prognostic or predictive character of the current identified relevant SNPs is not yet possible. An increased PAK function is often seen in tumours and is correlated with angiogenesis, tumour progression and a poor prognosis [36]. Yet, the influence of sunitinib and genetic variants on the PAK/PI3K/AKT pathway are not clear yet. Hypothetically, pharmacogenetic effects in this pathway could lead to drug resistance or a poor response to sunitinib [31,32,33,34,35,36,37,38].

The treatment arsenal for mRCC patients has undergone a very rapid development, with many new treatments being added in recent years. Immune checkpoint inhibitors (ICI) are now an essential element of therapy and there is increasing evidence that combination therapies with two ICIs or a ICI with a TKI are highly effective in mRCC [3]. Although sunitinib is no longer first-line treatment of mRCC, for some patients it still remains the best option or provides an effective second-line treatment for patients who have progressed to first-line ICIs. In addition, sunitinib now serves as a reference standard in studies to compare with new treatment options. It is therefore still important to be able to use sunitinib in the most optimal way. Indeed, while many treatment options for mRCC available, no validated markers have been identified yet to ensure that each individual receives the most appropriate treatment tailored to their individual characteristics. Our future aim therefore remains to use the patient’s genetic profile to predict the efficacy or toxicity on treatment with sunitinib, another TKI or an ICI. This GWAS serves as an important step in that direction. Likewise, it would be clinically relevant to investigate whether the variants in *PDLIM3* and *DSCAM* will also be predictive for PFS and OS of other TKIs given in mono- or combination therapy (e.g., axitinib and cabozantinib) to discover similar properties within this group of drugs. The knowledge gained from this GWAS and earlier candidate gene studies on sunitinib with regard to study design, data analysis and interpretation, will therefore be useful in follow-up studies on pharmacogenetics in other drugs used in mRCC or as a lead in new drug development. By building further on this research, we ultimately expect to know for each drug which pre-emptive genotyping test should be performed in order to optimally treat each unique mRCC patient.

The use of a structured eCRF ensured consistency in data collection by all participating EuroTARGET centres. Some single variables for calculating the Heng score were not present in all patients. Using the multiple imputation procedure to infer uncertain Heng risk group was necessary for only 17% of patients which we consider to be of low impact. The genome inflation factors in this analysis were above 1.05 suggesting slightly inflated test statistics. The moderate sample size prevented us from fitting larger models, for example including several principal components. Another limitation is the relatively small sample size for both studies, which substantially reduced the overlap of imputed SNPs when filtered at the 2% MAF level. Additionally, the small sample size leads to biased effect estimates when studying the strongest associated SNPs. HRs therefore have to be interpreted with care.

The findings presented in this study provide new insight into the pharmacogenetics of sunitinib with possible determinants of drug efficacy in mRCC patients (Appendix A).

## 5. Conclusions

We identified germline DNA variants in *PDLIM3* and *DSCAM* as novel determinants of PFS and OS in sunitinib-treated mRCC patients. The underlying link between the identified novel candidate genes and the molecular mechanisms of sunitinib action needs to be elucidated.

## Figures and Tables

**Figure 1 cancers-14-02838-f001:**
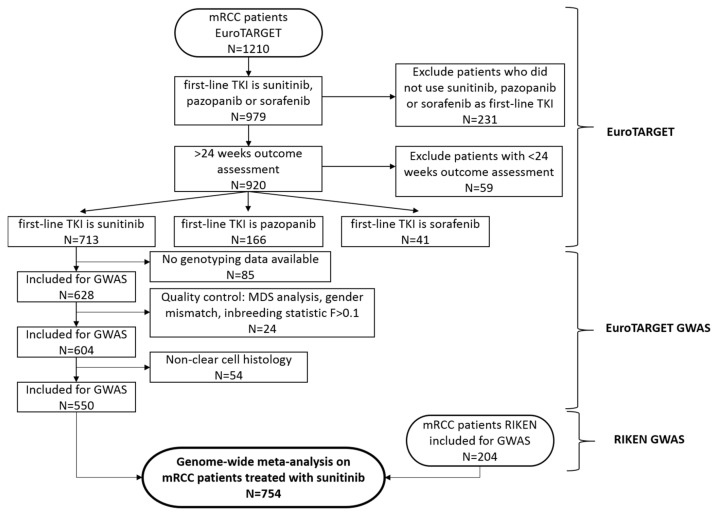
Flowchart on patient inclusion EuroTARGET and RIKEN. N indicates the number of patients. In total, 1210 mRCC patients were included in EuroTARGET, of which 748 were collected prospectively and 462 were available as historical (retrospective) series at the start of EuroTARGET [21]. Of the 1210 patients, we selected the 979 patients (81%) who received sunitinib, sorafenib, or pazopanib as first TKI (remainder of patients did for example have no treatment or were treated with an mTOR inhibitor or other TKI). We did not exclude patients who used cytokine therapy before the TKI. To allow for informative analyses, we only focused on the subset of 920 patients for whom outcome could be assessed for at least 6 months (24 weeks). Patients with missing genotypes (N = 85) were excluded for GWAS purposes. 24 patients were excluded after Quality Control (QC) checks. For efficacy analyses, 54 non-clear cell patients were excluded resulting in 550 patients available for the EuroTARGET GWAS [21]. An additional 204 patients from 15 Japanese medical institutes, the RIKEN cohort, was included to test efficacy endpoints in a genome-wide meta-analysis on mRCC patients treated with sunitinib.

**Figure 2 cancers-14-02838-f002:**
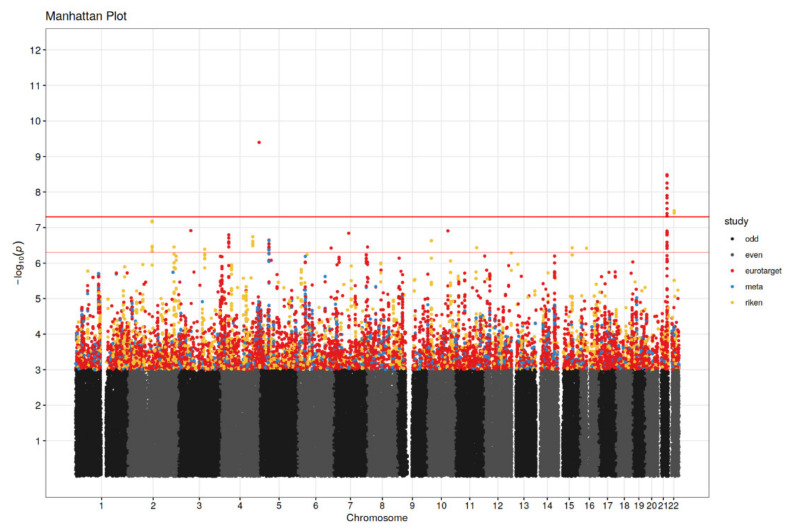
Manhattan plot for progression-free survival (PFS) analysis. Significance threshold (red horizontal line in bold) is set to *p* = 5 × 10^−8^ and *p*-values ≤ 5 × 10^−7^ (between red horizontal lines) were considered suggestive. Results are shown for EuroTARGET (red dots), RIKEN (green dots) and the meta-analysis (purple dots). Chromosomes are represented in odd numbers (black) and even numbers (grey).

**Figure 3 cancers-14-02838-f003:**
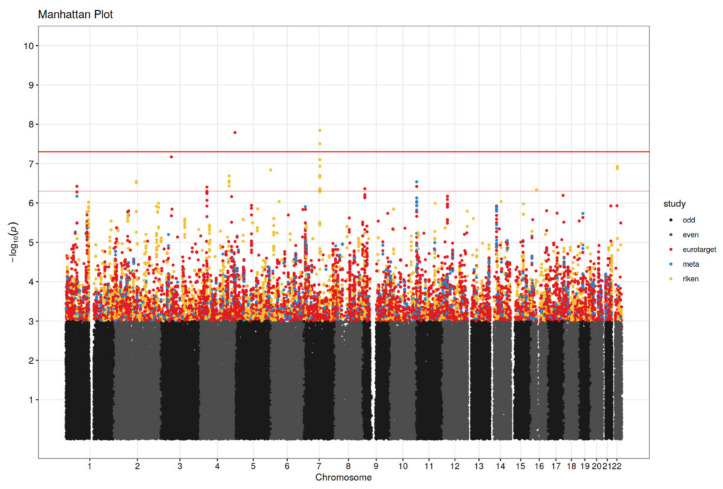
Manhattan plot for overall survival (OS) analysis. Significance threshold (red horizontal line in bold) is set to *p* = 5 × 10^−8^ and *p*-values ≤ 5 × 10^−7^ (between red horizontal lines) were considered suggestive. Results are shown for EuroTARGET (red dots), RIKEN (green dots) and the meta-analysis (purple dots). Chromosomes are represented in odd numbers (black) and even numbers (grey).

**Table 1 cancers-14-02838-t001:** Patient characteristics. Patients diagnosed with metastatic renal cell carcinoma, treated with sunitinib, and included for efficacy analyses for EuroTARGET cohort (*n* = 550) and Japanese RIKEN cohort (*n* = 204).

	EuroTARGET		RIKEN	
Characteristic	Value (Range)	%	Value (Range)	%
*Gender*				
■ male	405	74	151	74
■ female	145	26	53	26
Median age at start sunitinib (years)	63 (33–87)			
*Country*				
■ Netherlands	281	51		
■ Spain (SOGUG)	168	30.5		
■ Germany (CESAR)	39	7		
■ Iceland (Landspitali University Hospital)	27	5		
■ United Kingdom (Addenbrooke’s Cambridge)	20	3.5		
■ Romania (University of Medicine and Farmacy Carol Davila)	15	3		
■ Japan			204	100
*Prior nephrectomy before start sunitinib*				
■ yes	201	36.5	166	81
■ no	349	63.5	32	16
■ unknown			6	3
*WHO performance status (EuroTARGET), ECOG performance status (RIKEN)*				
■ 0	206	37.5	146	72
■ 1	206	37.5	44	22
■ 2	25	4.5	10	5
■ 3	2	0.3	2	1
■ 4	1	0.2	0	0
■ unknown	110	20	2	1
*Treatment line sunitinib*				
■ 1	512	93	98	48
■ 2	32	6	41	20
■ 3	5	0.9	39	19
■ 4	1	0.1	15	7
■ 5			4	2
■ Unknown			7	3
*Heng prognostic risk group* *				
■ Good (0 risk factors)	80	14.5		
■ Intermediate (1–2 risk factors)	298	54.2		
■ Poor (3–6 risk factors)	172	31.3		
*MSKCC prognostic risk group* **				
■ Favourable (0 risk factors)			38	19
■ Intermediate (1–2 risk factors)			124	61
■ Poor (3–6 risk factors)			32	16
■ Unknown			10	5
*Prior drugs (excluding TKI or TKI-like antitumor treatment)*				
■ Yes	38	7	102	50
■ No	512	93	95	47
■ Unknown			7	3
*Duration of prior drug treatment (months)*	1 to 23		1 to 123 months	
*Sunitinib starting dose (daily dose in mg)*				
■ 50	482	87.6	125	61
■ 37.5	46	8.4	72	35
■ 25	20	3.6	7	3
■ 12.5	2	0.4	0	0
*Median treatment duration of sunitinib*	10 months (2 days–113 months)		5 months (2 days–53 months)	
*Dose reduction*				
■ yes	223	41	0	0
■ no	310	56	201	99
■ unknown	17	3	3	1
*Dose reduction within 12 weeks (equal to 2 cycles) of treatment*				
■ yes	111	20	0	0
■ no	416	76	201	99
■ unknown	23	4	3	1
*Dose reduction within 24 weeks (equal to 4 cycles) of treatment*				
■ yes	145	26	0	0
■ no	382	70	201	99
■ unknown	23	4	3	1

Values are presented as median unless otherwise indicated. Abbreviations: WHO = World Health Organization. * The Heng prognostic risk group was based on six risk scores: WHO performance status (≥1), low haemoglobin (<lower limit of normal (LLN); for males LLN = 8.1 mmol/L OR 13 g/dL, for females LLN = 7.1 mmol/L OR 11.5 g/dL), high calcium (>2.5 mmol/L) and time from initial diagnosis to treatment with sunitinib (<1 year), neutrophil count (>upper limit of normal (ULN)) and thrombocytes (>ULN). The Heng risk groups as presented in this table were obtained after imputation of missing values. ** The MSKCC prognostic risk group was based on five risk scores: Karnofsky performance status (<80%), low haemoglobin (<lower limit of normal (LLN); for males LLN = 8.1 mmol/L OR 13 g/dL, for females LLN = 7.1 mmol/L OR 11.5 g/dL), high calcium (>2.5 mmol/L), high LDH (>1.5 × ULN), and time from initial diagnosis to treatment with sunitinib (<1 year).

**Table 2 cancers-14-02838-t002:** GWAS significantly or suggestively associated SNPs with PFS from the meta-analysis.

SNP	Chromosome	Position	Gene	MAF (%)	Allele *	*p*-Value	Hazard Ratio (HR)	Beta	Se	SOE	Association
rs28520013	4	186442067	*PDLIM3*	2.30	G/T	4.02 × 10^−10^	7.26	1.98	0.32	E	GWAS significant
rs2205096	21	41683405	*DSCAM*	6.30	T/A	5.60 × 10^−9^	2.50	0.92	0.16	E	
rs111356738	21	41677845	*DSCAM*	5.30	G/A	4.77 × 10^−8^	2.51	0.92	0.17	E	
rs76403021	3	58195886	*DNASE1L3*		G/C	1.22 × 10^−7^	4.23	1.44	0.27	E	GWAS suggestive
rs595883	10	98483774	Position close to *LOXL4*, *PYROXD2*, *HPS1*, and *HPSE2*	2.00	C/T	1.24 × 10^−7^	1.63	0.49	0.09	E	
rs118150161	7	71421614	*CALN1*	2.00	T/C	1.45 × 10^−7^	3.83	1.34	0.26	E	
rs113168647	21	41679847	*DSCAM*	2.00	C/T	1.59 × 10^−7^	2.46	0.90	0.17	E	
rs7282179	21	41670052	*DSCAM*	5.00	T/C	1.61 × 10^−7^	2.26	0.81	0.16	E	
rs79433348	4	41381011	*LIMCH1*	6.40	A/C	1.61 × 10^−7^	2.70	0.99	0.19	E	
rs79160607	4	41395370	*LIMCH1*	3.60	A/G	1.98 × 10^−7^	2.57	0.94	0.18	E	

* The first mentioned alleles are the reference alleles, the second mentioned alleles are the variant alleles. Abbreviations: DSCAM = Down Syndrome Cell Adhesion Molecule. MAF = Minor Allele Frequency. HR = Hazard Ratio. CI = Confidence Interval. SOE = source of evidence. E: Eurotarget, R: Riken, M: meta-analysis.

**Table 3 cancers-14-02838-t003:** GWAS significantly or suggestively associated SNPs with OS from the meta-analysis.

SNP	Chromosome	Position	Gene	MAF (%)	Allele *	*p*-Value	Hazard Ratio (HR)	Beta	Se	SOE	Association
rs28520013	4	186442067	*PDLIM3*	2.30	G	1.62 × 10^−8^	5.96	1.79	0.32	E	GWAS significant
rs62256189	3	55094589	*CACNA2D3*	2.50	G/T	6.75 × 10^−8^	3.20	1.16	0.22	E	GWAS suggestive
rs2271583	11	3685655	*ART1*	14.70	G/A	2.90 × 10^−7^	1.72	0.54	0.11	E	
rs80071112	1	57650640	*DAB1*	2.90	C/T	3.78 × 10^−7^	3.09	1.13	0.22	E	
rs79160607	4	41395370	*LIMCH1*	4.00	A/G	3.95 × 10^−7^	2.49	0.91	0.18	E	
rs10959526	9	11107610	Position close to *PTPRD*	4.20	T/C	4.36 × 10^−7^	2.52	0.93	0.18	E	

* The first mentioned alleles are the reference alleles, the second mentioned alleles are the variant alleles. Abbreviations: DSCAM = Down Syndrome Cell Adhesion Molecule. MAF = Minor Allele Frequency. OR = Odds Ratio. CI = Confidence Interval. SOE = source of evidence. E: Eurotarget, R: Riken, M: meta-analysis.

## Data Availability

All clinical and platform data generated in EuroTARGET are freely available in an anonymized way for the research community. The data can be accessed through the European Genome-phenome Archive (EGA) which is the controlled access repository under the European Bioinformatics Institute (EMBL-EBI). Interested parties will be able to find the EuroTARGET project under the Studies section.

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
