# Peer review of "Genome-Wide Meta-Analysis Identifies Variants in DSCAM and PDLIM3 That Correlate with Efficacy Outcomes in Metastatic Renal Cell Carcinoma Patients Treated with Sunitinib"

_cancers, 2022, doi:10.3390/cancers14122838_

Round 1
Reviewer 1 Report
Dear Editor, thank you so much for inviting me to revise this manuscript about renal cell carcinoma written by very important international experts.
This study addresses a current topic.
The manuscript is quite well written and organized.
Figures and tables are comprehensive and clear.
The introduction explains in a clear and coherent manner the background of this study.
We suggest the following modifications:
- Introduction section: although the authors correctly included important papers in this setting, we believe some recent studies regarding novel, emerging treatment options in this setting should be cited within the introduction (PMID: 31278395 ; PMID: 33714725), only for a matter of consistency. We think it might be useful to introduce the topic of this interesting study.
- Methods and Statistical Analysis: nothing to add.
- Discussion section: Very interesting and timely discussion. Of note, the authors should expand the Discussion section, including a more personal perspective to reflect on. For example, they could answer the following questions – in order to facilitate the understanding of this complex topic to readers: what potential does this study hold? What are the knowledge gaps and how do researchers tackle them? How do you see this area unfolding in the next 5 years? We think it would be extremely interesting for the readers.
However, we think the authors should be acknowledged for their work. In fact, they correctly addressed an important topic in renal cell carcinoma, the methods sound good and their discussion is well balanced.
One additional little flaw: the authors could better explain the limitations of their work, in the last part of the Discussion.
We believe this article is suitable for publication in the journal although some revisions are needed. The main strengths of this paper are that it addresses an interesting and very timely question and provides a clear answer, with some limitations.
We suggest the addition of some references for a matter of consistency. Moreover, the authors should better clarify some points.
Reviewer 2 Report
Τhe authors describe two new genes with their alterations being of cli
nical relevance in RCC. Overall this is an interesting work however there is a main limitation that the authors
I qdo not mention which is that sunitinib is now obsolete as a first line therapy for metastayic RCC in view of currrnt immunotherapy TKI combination approvals. This wouls significantly limit the clinical significance of this study's findings. Can the authors propose hiw their resukts could be potentially integrated in the current treatment armamentarium and prognostigation models for mESS
Includong IMDC, MSKCC and MDACC criteria?
Finally there is aome english polishing that needs to be addressed throughout the manuscript
Reviewer 3 Report
Diekstra and colleagues report a "genome-wide meta-analysis [that] identifies variants in DSCAM and PDLIM3 that correlate with efficacy outcomes in metastatic renal cell carcinoma patients treated with sunitinib". This manuscript represents an interesting study, but will require more solid data to support the conclusions. The main concerns are as follows:
- The significance of this association study was established arbitrarily at P<5·10-8 without any correction for multiple testing. Thus, a correction for multiple testing, such as Bonferroni or Benjamini and Hochberg, should be used in the study to take into account the hundred of thousands markers evaluated with the Illumina Human OmniExpress BeadChip arrays. Otherwise the results might be misleading.
- The Cox regression of SNPs were not compared to clinical variables. Thus, the authors have to perform a multivariate Cox regression using a forward and/or backward stepwise conditional Cox regression model including clinical variables, such as grade, T, M, N and stage. The variables that remain in that model can then be considered as independent prognostic factors of PFS or overall survival.
- To have clinical significance, results should be validated with additional large external datasets, such as KIRC-TCGA, which contain >500 patient samples.
Reviewer 4 Report
In this study, the author identified some variants which are correlated with outcome of sunitinib. There are some concerns. Please check the following points.
- In figure 1, according to the flowchart, “first line TKI is sunitinib” group was selected. But, in Table 1, treatment line sunitinib is not only first line. The title indicates that some variants the author identified has a great impact on the outcome of sunitinib treatment. But, these cohorts included several TKI treatments.
- Histology and clinical information is important for TKI treatment. Please show them.
Round 2
Reviewer 1 Report
Acceptance.
Reviewer 2 Report
No additional comments.
Reviewer 3 Report
The authors clarified some of the concerns raised by this reviewer, but there are still issues that need to be addressed before publication:
- The SNPs associated with PFS and OS in Tables 2 and 3 need to be placed in context with clinical variables known to be strongly associated with PFS and OS, such as grade, T, M, N and stage in multivariate Cox analyses. It is recommended a forward and/or backward stepwise conditional Cox regression model in order to claim that SNPs are independent prognostic factors of PFS and OS.
- The results need to be validated with additional large independent datasets, such as KIRC-TCGA, which contain the genotyping, treatments and clinical variables of >500 renal cell carcinoma patient samples, or datasets from other studies.
- All genotyping arrays from all samples must be deposited in Gene Expression Omnibus (GEO) or ArrayExpress public databases.
Reviewer 4 Report
Thank the author for the revising the manuscript.
- I understand that the clinical information was described in the previous study. But, in this study, excluded some patients from Euro TARGET. In addition, the RIKEN cohort is also used in this study. Please add the clinical information such as TNM to the Table 1.
Round 3
Reviewer 4 Report
The manuscript is well revised.